# Chromosomes of Asian cyprinid fishes: Novel insight into the chromosomal evolution of Labeoninae (Teleostei, Cyprinidae)

**Sudarat Khensuwan**[1]☯, **Francisco de Menezes Cavalcante Sassi**[2]☯, **Renata Luiza Rosa de Moraes**[2], **Petr Rab**[3], **Thomas Liehr**[4], **Weerayuth Supiwong**[5]*, **Kriengkrai Seetapan**[6], **Alongklod Tanomtong**[1], **Nathpapat Tantisuwichwong**[1], **Satit Arunsang**[7], **Phichaya Buasriyot**[8], **Sampun Tongnunui**[9], **Marcelo de Bello Cioffi**[2]

1 Department of Biology, Faculty of Science, Khon Kaen University, Muang, Khon Kaen, Thailand, 2 Departamento de Genética e Evolução, Laboratório de Citogenética de Peixes, Universidade Federal de São Carlos, São Carlos, São Paulo, Brazil, 3 Institute of Animal Physiology and Genetics, Laboratory of Fish Genetics, Czech Academy of Sciences, Rumburská, Liběchov, Czech Republic, 4 Institute of Human Genetics, University Hospital Jena, Jena, Germany, 5 Faculty of Interdisciplinary Studies, Khon Kaen University, Nong Khai Campus, Muang, Nong Khai, Thailand, 6 School of Agriculture and Natural Resources, University of Phayao, Tumbol Maeka, Muang, Phayao, Thailand, 7 Program in Animal Science, Faculty of Agricultural Technology and Agro-Industry, Rajamangala University of Technology Suvarnabhumi, *Phra Nakhon* Si Ayutthaya, Ayutthaya, Thailand, 8 Faculty of Science and Technology, Rajamangala University of Technology Suvarnabhumi, Mueang Nonthaburi, Nonthaburi, Thailand, 9 Department of Conservation Biology, Mahidol University, Kanchanburi Campus, Sai Yok, Kanchanaburi Province, Thailand

☯ These authors contributed equally to this work.
* weersu@kku.ac.th

**Data Availability Statement:** All relevant data are within the paper.

## Abstract

The Labeoninae subfamily is a highly diversified but demonstrably monophyletic lineage of cyprinid fishes comprising five tribes and six incertae sedis genera. This widely distributed assemblage contains some 48 genera and around 480 recognized species distributed in freshwaters of Africa and Asia. In this study, the karyotypes and other chromosomal properties of five Labeoninae species found in Thailand *Labeo chrysophekadion* (Labeonini) and *Epalzeorhynchos bicolor*, *Epalzeorhynchos munense*, *Henicorhynchus siamensis*, *Thynnichthys thynnoides* (´Osteochilini´) were examined using conventional and molecular cytogenetic protocols. Our results confirmed a diploid chromosome number (2n) invariably 2n = 50, but the ratio of uni- and bi-armed chromosomes was highly variable among their karyotypes, indicating extensive structural chromosomal rearrangements. Karyotype of *L. chrysophekadion* contained 10m+6sm+20st+14a, 32m+10sm+8st for *H. siamensis*, 20m+12sm+10st+8a in *E. bicolor*, 20m+8sm+8st+14a in *E. munense*, and 18m+24sm+8st in *T. thynnoides*. Except for *H. siamensis*, which had four sites of 5S rDNA sites, other species under study had only one chromosome pair with those sites. In contrast, only one pair containing 18S rDNA sites were found in the karyotypes of three species, whereas two sites were found in that of *E. bicolor*. These cytogenetic patterns indicated that the cytogenomic divergence patterns of these labeonine species largely corresponded to the inferred phylogenetic tree. In spite of the 2n stability, diverse patterns of rDNA and microsatellite distribution as well as their various karyotype structures demonstrated significant evolutionary differentiation of Labeoninae genomes as exemplified in examined species. Labeoninae offers a

**Funding:** This research was supported by the National Research Council of Thailand (NCRT) in the form of a grant to SK [NRCT-RGJ63003-068] and by the Thailand science research and innovation fund and the University of Phayao in the form of a grant to KS [FF66-UoE013]. The funders had no role in study design, data collection and analysis, decision to publish, or preparation of the manuscript."

**Competing interests:** The authors have declared that no competing interests exist.

traditional point of view on the evolutionary forces fostering biological diversity, and the recent findings add new pieces to comprehend the function of structural chromosomal rearrangements in adaption and speciation.

## Introduction

Around 48 genera and more than 480 species compose the demonstrably monophyletic Labeoninae (Cyprinidae, Cypriniformes) [1], which inhabits the fast-flowing rivers and streams of Africa and Asia [2]. Because of their beautiful body patterns and large size of some species, these fishes are frequently targets of the aquarium trade as well as artisanal fisheries, accounting for 18% of total aquatic animal production in 2020 [3]. However, most cytogenetic research in this group is limited to the report of the diploid chromosome number (2n) and the karyotype composition [4, 5], which does not preclude recognition of a wide range of chromosomal diversity.

All Labeoninae species are evolutionary diploid, with 2n ranging from 48 to 50, in contrast to the tetra- and hexaploid lineages of other Cyprinidae subfamilies, namely Barbinae, Cyprininae, Probarbinae, Schizopygopsinae, Schizothoracinae, Smiliogastrinae, Spinibarbinae, and Torinae [1, 4, 6, 7]. Phylogenetic reconstructions suggest that the divergence of cyprinid lineages began with diploid species about 70 million years ago (Mya), progressed through a tetraploidization stage between 45 and 20 Mya, and then experienced a more recent hexaploidization event in the last 20 Mya. However, the exact timing of this divergence is still debated [7]. According to Yang et al. [7], the Labeoninae family underwent diversification around 43.6 Mya and was found to be the sister basal branch of all other cyprinids [6–9].

The species of the evolutionary diploid subfamilies Acrossocheilinae and 'Poropuntiinae' both have a conserved 2n of 50, however those of Labeoninae and Smiliogastrinae vary between values of 48 and 50 chromosomes (Table 1). This reduction in 2n was most likely caused by chromosomal fusions, which, in certain rare cases, are associated with the origin of the sex chromosomes [10]. Despite the widespread conservation of their 2n, Labeoninae species have differences in the macro- and microstructure of the karyotypes as evidenced by the different fundamental numbers (NF) and 18S rDNA sites distribution (Table 1). This suggests that pericentric inversions played a significant role in their karyotype evolution. In this context, molecular cytogenetics emerges as a crucial tool to identify the karyotype dynamics in fishes, as has already been demonstrated for several Cyprinidae species [11–18]. Indeed, the mapping of repetitive sequences, which make up the majority of the eukaryotic genome, is widely utilized in fish cytogenetics and has a significant potential to further understand the origin process of sex and B chromosomes, as well as karyotype differentiation [19].

In general, available data demonstrate that although almost all cyprinoid species share the same macrostructure of their karyotypes (i.e.: 2n = 50 and karyotypes dominated by bi-armed chromosomes), they diverge in patterns of C-positive heterochromatin, and rDNA distribution [18, 20–26]. Considering this scenario, our study provides novel cytogenetic data for five Labeoninae species, namely *Epalzeorhynchos bicolor* Smith (1931), *E. munense* Smith (1934), *Henicorhynchus siamensis* Sauvage (1881), *Thynnichthys thynnoides* Bleeker (1852) and *Labeo chrysophekadion* Bleeker (1849) from two tribes (Osteochilini and Labeonini *sensu* Tan & Armbruster, 2018). These results add to our understanding of the ways in which structural chromosomal rearrangements play a role in speciation and adaptation.

**Table 1. Review of cytogenetic data in the tribe Labeoninae, excluding *incertae sedis* genera (sensu Tan and Armbruster [1]).** The species analyzed in this study are highlighted in red.

| Species | 2n | NF | Karyotype | NORs/18S rDNA pairs | References |
|---|---|---|---|---|---|
| *Bangana devdevi* | 50 | 86 | 20m+16sm+14a | - | [27] |
| *Barbichthys laevis* | 50 | 76 | 20m+6sm+14a | - | [28] |
| *Cirrhinus julleini* | 50 | 90 | 26m+14sm+4st+6a | - | [29] |
| *Cirrhinus microlepis* | 50 | 72 | 12m+10sm+2st+26a | - | [30] |
| *Cirrhinus molitorella* | 50 | 98 | 20m+26sm+2st+2a | - | [31] |
| *Cirrhinus molitorella* | 50 | 100 | 16m+24sm+10st | four pairs | [32–35] |
| *Cirrhinus mrigala* | 50 | 82 | 10m+12sm+10st+18a | two pairs | [29] |
| *Cirrhinus mrigala* | 50 | 100 | 4m+18sm+28st | - | [36] |
| *Cirrhinus mrigala* | 50 | 78 | 8m+6sm+14st+22a | - | [37] |
| *Cirrhinus mrigala* | 50 | 86 | 12m+18sm+6st+14a | two pairs | [38] |
| *Cirrhinus reba* | 50 | 78 | 6m+8sm+14st+22a | - | [37] |
| *Cirrhinus reba* | 50 | 92 | 18m+20sm+6st+4a | - | [39] |
| *Crossocheilus latius latius* | 50 | 90 | 12m+28sm+10a | - | [40, 41] |
| *Crossocheilus latius latius* | 50 | 82 | 8m+12sm+12st+18a | - | [42] |
| *Crossocheilus latius punjabensis* | 48 | 60 | 12m+36a | - | [43] |
| *Discogobio tetrabarbatus* | 50 | 90 | 10m+18sm+12st+10a | - | [32, 33, 35] |
| *Epalzeorhynchos bicolor* | 50 | 52 | 20m+4sm+2st+24a | - | [44] |
| *Epalzeorhynchos bicolor* | 50 | 84 | 20m+12sm+10st+8a | 19, 21 | Present study |
| *Epalzeorhynchos frenatum* | 48 | 72 | 14m+10sm+8st+16a | - | [44] |
| *Epalzeorhynchos frenatum* | 50 | 76 | 18m+10sm+10st+12a | 10 | [5] |
| *Epalzeorhynchos frenatum* | 50 | 72 | 20m+8sm+8st+14a | 11 | Present study |
| *Epalzeorhynchos munensis* | 50 | 84 | 22m+12sm+2st+14a | - | [45] |
| *Garra cambodgiensis* | 50 | 82 | 20m+12sm+4st+14a | - | [46] |
| *Garra cambodgiensis* | 50 | 76 | 8m+18sm/st+24a | - | [47, 48] |
| *Garra dembeensis* | 50 | 82 | 14m+18sm+18a | - | [49] |
| *Garra fasciacauda* | 50 | 84 | 18m+14sm+2st+16a | - | [46] |
| *Garra gotyla gotyla* | 50 | 84 | 14m+10sm+10st+16a | - | [50] |
| *Garra gotyla gotyla* | 50 | 90 | 14m+26sm+10a | - | [51] |
| *Garra gotyla gotyla* | 50 | 84 | 12m+8sm+8st+22a | - | [52] |
| *Garra imberba* | 50 | 96 | 14m+20sm+12st+4a | - | [53, 35] |
| *Garra kempi* | 50 | 88 | 14m+14sm+10st+12a | - | [52] |
| *Garra lamta* | 50 | 88 | 12m+24sm+2st+12a | - | [54] |
| *Garra lamta* | 50 | 86 | 6m+18sm+12st+14a | - | [55] |
| *Garra lissorhynchus* | 50 | 88 | 16m+16sm+6st+12a | - | [52] |
| *Garra makiensis* | 50 | 84 | 14m+20sm+16a | - | [49] |
| *Garra mullya* | 50 | 92 | 18m+14sm+10st+8a | two pairs | [56] |
| *Garra notata* | 50 | 80 | 20m+10sm+20a | - | [46] |
| *Garra orientalis* | 50 | 92 | 16m+12sm+14st+8a | - | [32, 33,35] |
| *Garra ornata* | 50 | - | - | - | [57] |
| *Garra quadrimaculata* | 50 | 88 | 16m+22sm+12a | - | [49] |
| *Garra rufa obtusa* | 44–52 | - | - | - | [58] |
| *Garra rufa rufa* | 44–52 | - | - | - | [58] |
| *Garra rufa* | 44 | 86 | 22m+20sm+2a | - | [52] |
| *Garra surendranathanii* | 50 | 92 | 14m+20sm+8st+8a | - | [56] |
| *Garra trewavasae* | 50 | 65 | 10m+3sm+2st+35a | - | [59] |
| *Garra variabilis* | 74 | - | - | - | [58] |

*(Continued)*

**Table 1.** (Continued)

| Species | 2n | NF | Karyotype | NORs/18S rDNA pairs | References |
|---------|-----|-----|-----------|---------------------|------------|
| *Henicorhynchus siamensis* | 50 | 100 | 32m+10sm+8st | 18 | Present study |
| *Labeo alluaudi* | 50 | - | - | - | [57] |
| *Labeo bata* | 50 | 80 | 18m+12sm+8st+12a | two pairs | [60] |
| *Labeo bata* | 50 | 74 | 6m+18sm+16st+10a | - | [39] |
| *Labeo behri* | 50 | 70 | 12m+8sm+2st+28a | - | [61] |
| *Labeo calbasu* | 50 | 70 | 10m+10sm+14st+16a | two pairs | [60] |
| *Labeo calbasu* | 50 | 64 | 6m+8sm+22st+14a | - | [39] |
| *Labeo calbasu* | 50 | 80 | 8m+22sm/st+20a | two pairs | [62, 63] |
| *Labeo caeruleus* | 48 | 66 | 12m+6sm+6st+24a | - | [37] |
| *Labeo chrysophekadion* | 50 | 78 | 4m+10sm+14st+22a | - | [64] |
| *Labeo chrysophekadion* | 50 | 72 | 10m+6sm+20st+14a | 12,13 | [65] |
| *Labeo chrysophekadion* | 50 | 72 | 10m+6sm+20st+14a | 9 | Present study |
| *Labeo coubie* | 50 | - | - | - | [66] |
| *Labeo dero* | 50 | 88 | 26m+12sm+2st+10a | - | [67] |
| *Labeo dero* | 48 | 76 | 12m+16sm+20st/a | - | [68] |
| *Labeo diplostomus* | 50 | 66 | 10m+6sm+8st+26a | - | [50] |
| *Labeo dussumieri* | 50 | 74 | 12m+12sm+10st+16a | four pairs | [69] |
| *Labeo gonius* | 54 | 54 | 54a | two pairs | [43] |
| *Labeo pangusia* | 50 | 68 | 6m+12sm+16st+16a | - | [42] |
| *Labeo parvus* | 50 | - | - | - | [57] |
| *Labeo parvus* | 50 | - | - | - | [57] |
| *Labeo rohita* | 50 | 80 | 14m+16sm+8st+12a | two pairs | [29] |
| *Labeo rohita* | 50 | 74 | 10m+14sm+6st+20a | two pairs | [60] |
| *Labeo rohita* | 50 | 80 | 10m+20sm+20a | - | [36] |
| *Labeo rohita* | 50 | 70 | 8m+12sm+6st+24a | - | [70] |
| *Labeo rohita* | 50 | 74 | 10m+14sm+8st+18a | four pairs | [38] |
| *Labeo rohita* | 50 | 88 | 10m+16sm+12st+12a | - | [32, 33, 35] |
| *Labeo roseopunctatus* | 50 | - | - | - | [66] |
| *Labeo rouaneti* | 50 | - | - | - | [57] |
| *Labeo senegalensis* | 50 | - | - | - | [66] |
| *Labiobarbus lineatus* | 50 | 80 | 20m+10st+20a | - | [71] |
| *Labiobarbus leptocheilus* | 50 | 86 | 14m+6sm+16st+14a | 2 | [17] |
| *Lobocheilos rhabdoura* | 50 | 96 | 10m+18sm+12st+10a | 6 | [26] |
| *Osteochilus hasselti* | 50 | 96 | 30m+14sm+6st | - | [71] |
| *Osteochilus lini* | 50 | 100 | 12m+34sm+4st | 12 | [18] |
| *Osteochilus melanopleura* | 50 | 96 | 22m+24sm+2st+2a | 2 | [18] |
| *Osteochilus microcephalus* | 50 | 100 | 14m+32sm+4st | 3 | [18] |
| *Osteochilus vittatus* | 50 | 96 | 16m+30sm+4st | 12 | [18, 71] |
| *Osteochilus waandersi* | 50 | 92 | 18m+24sm+4st+4a | 15 | [18, 29] |
| *Parasinilabeo assimilis* | 50 | 96 | 16m+12sm+18st+4a | - | [32, 35] |
| *Semilabeo notabilis* | 50 | 80 | 8m+10sm+12st+20a | - | [32, 35] |
| *Semilabeo prochilus* | 50 | 98 | 16m+18sm+14st+2a | - | [35, 53] |
| *Semilabeo prochilus* | 50 | 100 | 16m+20sm+14st | seven pairs | [72] |
| *Thynnichthys thynnoides* | 50 | 100 | 18m+24sm+8st | 10 | Present study |

## Material and methods

### Sampling

Samples have been collected at five distinct sites that belong to Thailand's Mae Klong, Songkhram, and Chao Phraya drainage systems, as indicated in Fig 1 and Table 2. We included two species that were collected in the same basins, but were very low abundance in wild habitats [73, 74]. Considering that bloodstock wilds have been successfully produced by artificial insemination in aquaculture farms in Thailand, fish samples were also purchased directly from the

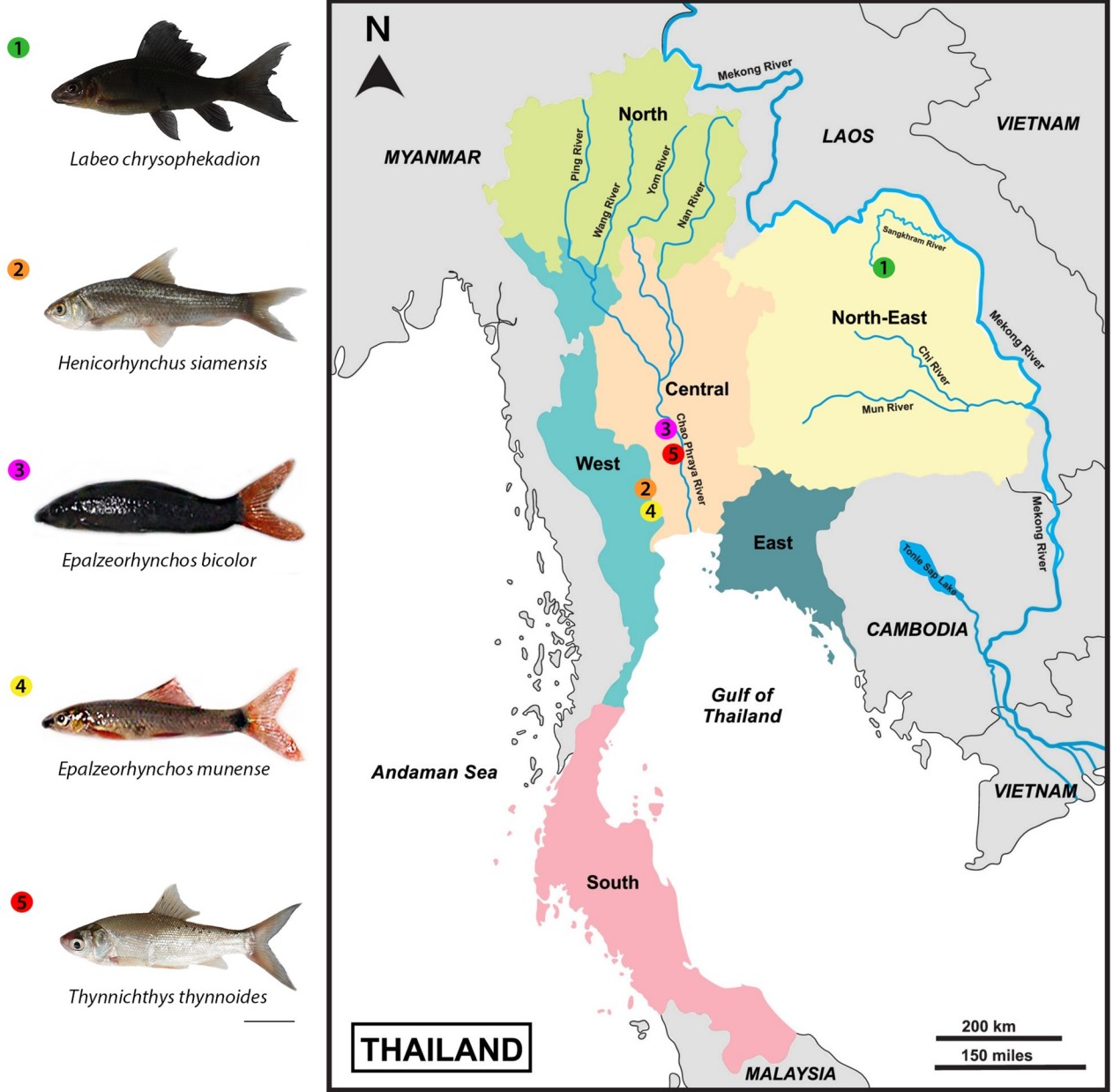

**Fig 1. Distribution of five Labeoninae species here investigated.** Thailand map showing the collection sites of the five species studied including 1. *Labeo chrysophekadion* (green circles); 2. *Henicorhynchus siamensis* (orange circles); 3. *Epalzeorhynchos bicolor* (pink circles); 4. *Epalzeorhynchos munense* (yellow circles); 5. *Thynnichthys thynnoides* (red circles). Scale bar = 1 cm.

**Table 2. The collection sites of the five Labeoninae species and the number of analyzed individuals (n) per sex.**
Individuals of *E. bicolor* and *E. munense* were gathered from Ban Pong Fish Market, Ratchaburi province (13˚ 51'42.6"N 99˚52'48.8"E).

| Species | Locality | n |
|---|---|---|
| *Epalzeorhynchos bicolor* | Chao Phraya Basin (site 1) | 07♀; 06♂ |
| *Epalzeorhynchos munense* | Mae Klong Basin (site 2) | 05♀; 06♂ |
| *Henicorhynchus siamensis* | Mae Klong Basin (site 3) | 09♀; 10♂ |
| *Labeo chrysophekadion* | Songkhram Basin (site 4) | 06♀; 08♂ |
| *Thynnichthys thynnoides* | Chao Phraya Basin (site 5) | 07♀; 09♂ |

Ban Pong Fish Market in Ratchaburi province. The analyzed individuals have been deposited in the fish collections at KhonKaen University's Cytogenetic Laboratory, Department of Biology, and Faculty of Science. All the species examined here were correctly identified using the morphological criteria [75, 76]. Mitotic chromosomes were obtained from the anterior kidney according to Bertollo et al. [77]. We used silver nitrate ($AgNO_3$) to visualize the nucleolar organizer regions (Ag-NOR) [78]. We used clove oil (Eugenol 3%) for anesthesia following the sacrifice of the individuals, as approved by the Animal Ethics Committee of KhonKaen University based on the Ethics of Animal Experimentation of the National Research Council of Thailand (Record No. IACUC-KKU-105/63). This research also received authorization to the animal care and use license for scientific research U1-07864-2561 (COA.No.MU-IACUC 2019/007). For the fishes of the Mae Klong Basin, study was approved in Mahidol University and fish collection authorized by Department of fisheries in Thailand, license number was 49/2564.

## Fluorescence in situ hybridization (FISH)

Fluorescence *in situ* hybridization experiments were conducted under high stringency conditions [79] to identify both 5S and 18S ribosomal DNA classes and the microsatellites $(CA)_{15}$, $(GC)_{15}$, $(TA)_{15}$, $(CAA)_{10}$ $(GAA)_{10}$ $(CAG)_{10}$ $(CAT)_{10}$ and $(CGG)_{10}$ The 5S ribosomal probe had 200 bp of the non-transcribed spacer (NTS) and 120 base pairs (bp) of the 5S rRNA transcribing gene [80]. The 18S rDNA was derived from the nuclear DNA of the wolf fish *Hoplias malabaricus* using PCR, and it corresponded to the 1400 bp segment of the 18S rRNA gene [81]. Both rDNA probes were directly labelled with the Nick-Translation mix kit (Jena Bioscience, Jena, Germany), where 5S rDNA was labeled in red with Atto550-dUTP and the 18S rDNA was labelled in green with Atto448-dUTP, according to the manufacturer's instructions. The microsatellite sequences were directly labelled with Cy-3 during the synthesis, described by Kubat et al. [82].

## Image processing

To confirm the results, we used at least 20 metaphase spreads per individual. Images were captured with an Axioplan II microscope (Carl Zeiss Jena GmbH, Germany) with CoolSNAP, and processed using ISIS (MetaSystems Hard & Software GmbH). Chromosomes were classified as metacentric (m), submetacentric (sm), subtelocentric (st), and acrocentric (a) according to their short/long arms ratio [83].

## Results

All five examined species, for both females and males, possessed invariably, 2n = 50, but with distinct karyotype compositions: 10m+6sm+20st+14a in *Labeo chrysophekadion*, 32m+10sm

+8st in *Henicorhynchos siamensis* (Fig 2A and 2B), 20m+12sm+10st+8a in *Epalzeorhynchos bicolor*, 20m+8sm+8st+14a in *E. munense*, and 18m+24sm+8st in *Thynnichthys thynnoides* (Fig 3A, 3B and 3G). Single Ag-NOR sites were detected in all species at the telomeric region of *p* arms. In the genome of *E. bicolor* and *E. munense*, the 5S rDNA was distributed in the

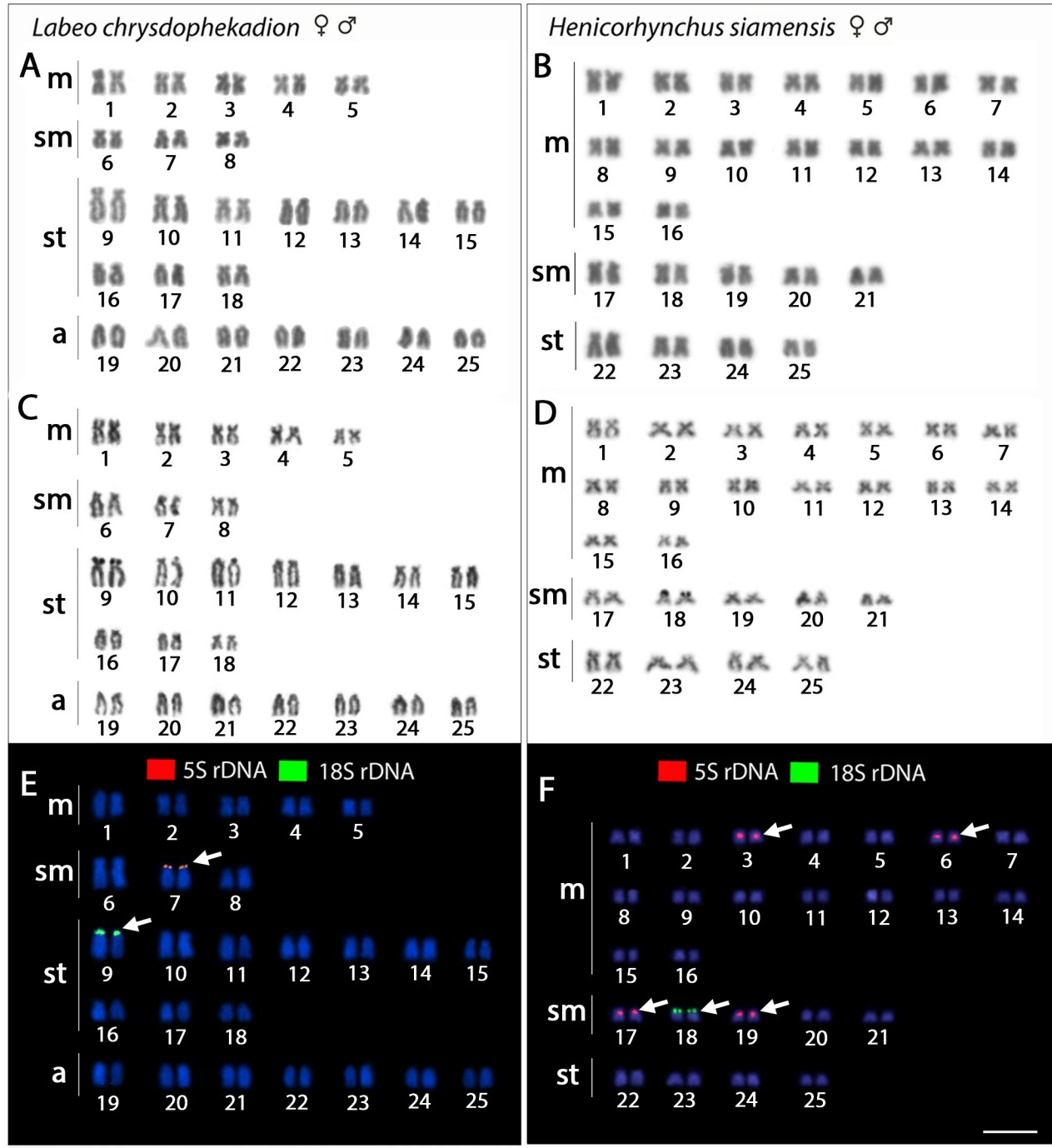

**Fig 2. Karyotypes of *Labeo chrysophekadion* and *Henicorhyncus siamensis*.** Karyotypes of *L. chrysophekadion* (A,C,E) and *Henicorhyncus siamensis* (B,D,F) organized from chromosomes after Giemsa-staining, Ag-NOR impregnation and FISH with 5S and 18S probes. Chromosomes were counterstained with DAPI (blue). Arrows indicate the positive FISH results. Bar = 5 μm.

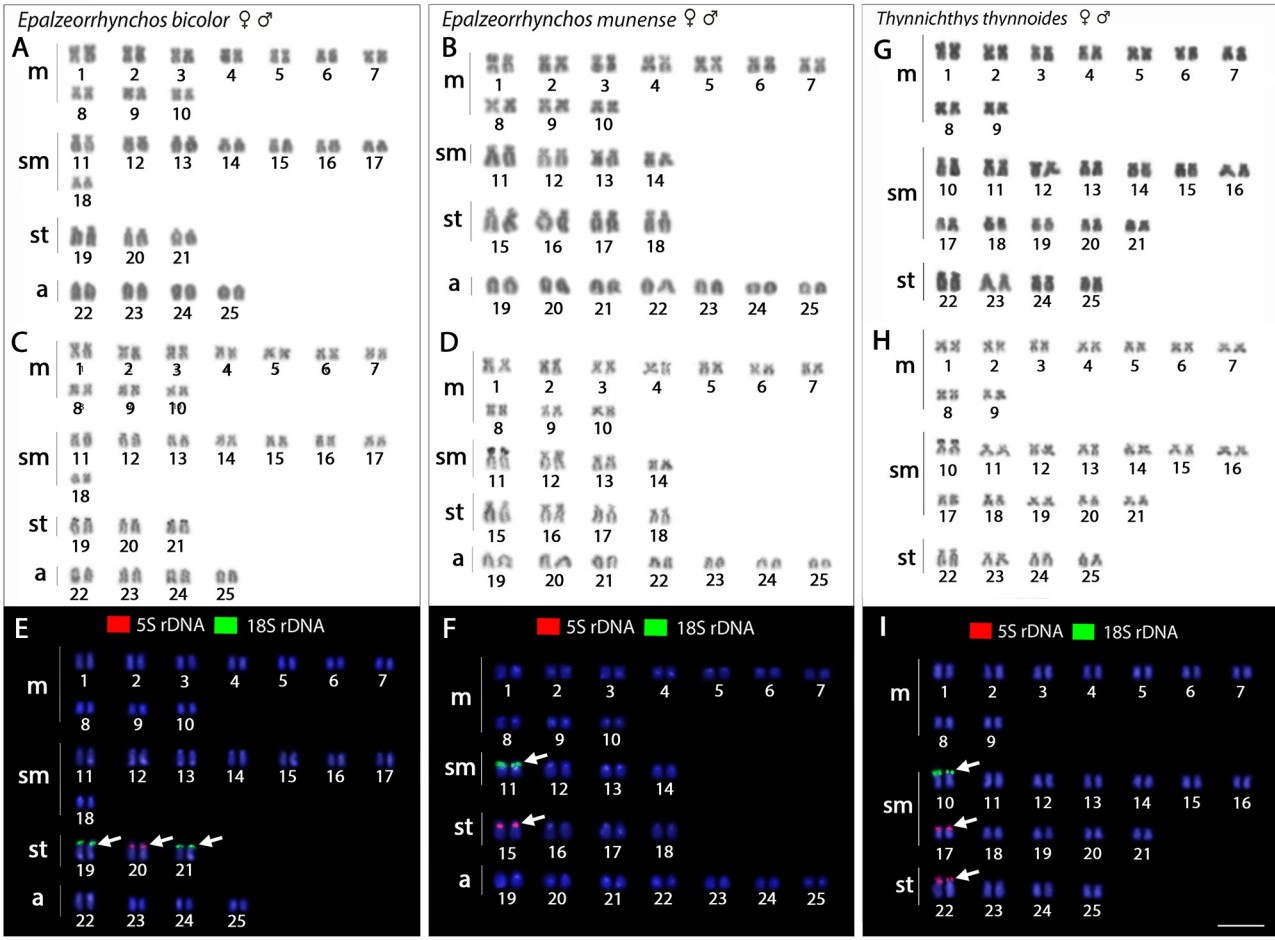

**Fig 3. Karyotypes of *Epalzeorrhyncos bicolor*, *E. munense*, and *Thynnichthys thynnoides*.** Karyotypes of *E. bicolor* (A,C,E), *E. munense* (B,D,F), and *T. thynnoides* (G,H,I) arranged from Giemsa- and Ag-NOR-stained chromosomes and chromosomes after FISH with 5S (red) and 18S (green) rDNA probes. Chromosomes were counterstained with DAPI (blue). The positive FISH signals are indicated by arrows. Scale bar = 5 μm.

centromeric region of st pairs 20 and 15. On the other hand, *H. siamensis* had these sites located in four chromosomal pairs at the centromeric region (pairs 3, 6, 17, 19). For *L. chryso-phekadion* (pair 7, Fig 2E) and *T. thynnoides* (pairs 17, 22, Fig 3I), the 5S rDNA was found in the telomeric region of the p arms. The 18S rDNA distribution corresponded to the Ag-NOR signals: pair 9 in *L. chrysophekadion*, pair 18 in *H. siamensis* pairs 19 and 21 in *E. bicolor*, pair 11 in *E. munense*, and pair 10 in *T. thynnoides*.

Microsatellite mapping showed species-specific patterns and mostly telomeric signals for all species (Figs 4–8). For the $(CA)_{15}$, dispersed signals were seen in the chromosomes of *L. chry-sophekadion*, whereas *T. thynnoides* and *H. siamensis* only displayed telomeric signals, in con-trast to *E. bicolor* and *E. munense*, which additionally display one set of signals marked in the pericentromeric region. The $(GC)_{15}$ was found in the telomeres of *T. thynnoide*s and *E. munense*; however, in *E. bicolor* and *H. siamensis*, only one chromosome pair was marked, and this motif was dispersed in *L. chrysophekadion*. The $(GAA)_{10}$ accumulates in the telomeres of other species but was distributed in the chromosomes of *T. thynnoides* and *E. munense*. On the other hand, $(CAT)_{10}$ was only accumulated in telomeres of *E. bicolor*, in contrast to dispersed in other species. All species, except *E. munense* with a single chromosome pair marked, had

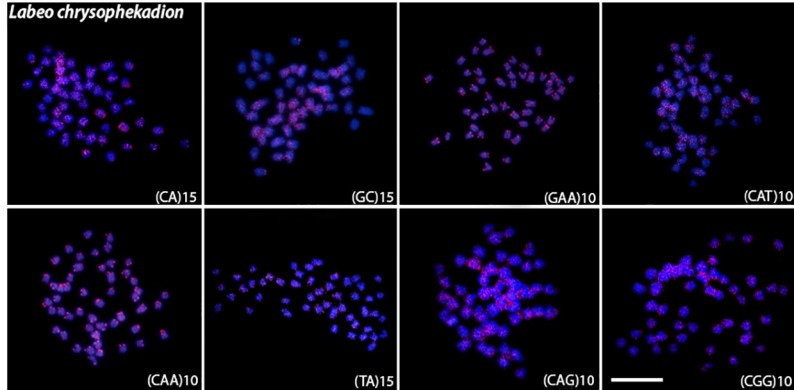

**Fig 4. Microsatellite sequences mapped onto *L. chrysophekadion* chromosomes.** Distribution of microsatellite sequences $(CA)_{15}$, $(GC)_{15}$, $(TA)_{15}$, $(CAA)_{10}$ $(GAA)_{10}$ $(CAG)_{10}$ $(CAT)_{10}$, and $(CGG)_{10}$ onto *L. chrysophekadion* metaphase chromosomes.

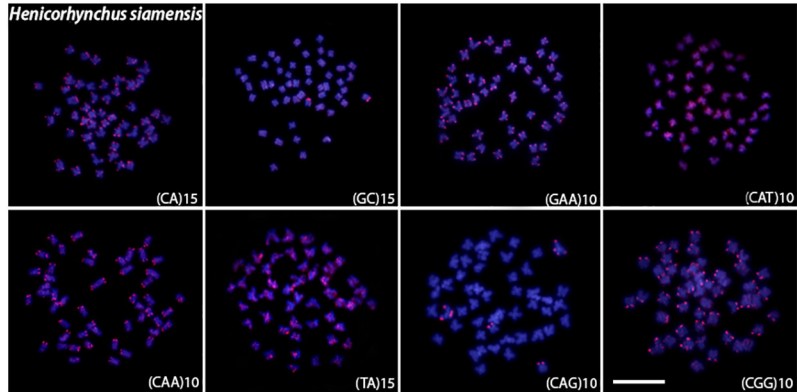

**Fig 5. Microsatellite sequences mapped onto *H. siamensis* chromosomes.** Distribution of microsatellite sequences $(CA)_{15}$, $(GC)_{15}$, $(TA)_{15}$, $(CAA)_{10}$ $(GAA)_{10}$ $(CAG)_{10}$ $(CAT)_{10}$ and $(CGG)_{10}$ onto *H. siamensis* metaphase chromosomes.

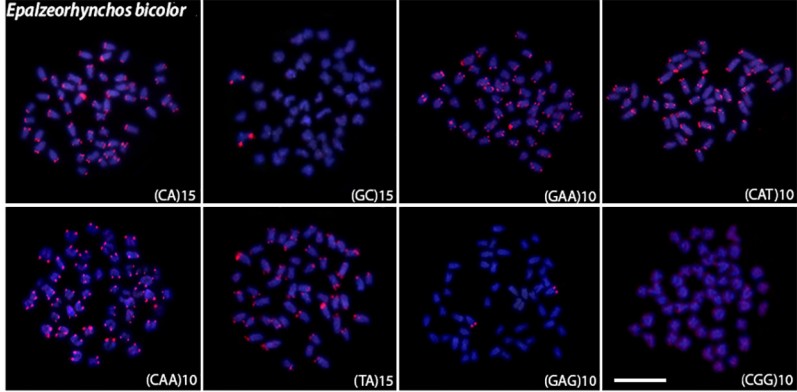

**Fig 6. Microsatellites sequences mapped onto *E. bicolor* chromosomes.** Distribution of microsatellite sequences $(CA)_{15}$, $(GC)_{15}$, $(TA)_{15}$, $(CAA)_{10}$ $(GAA)_{10}$ $(CAG)_{10}$ $(CAT)_{10}$ and $(CGG)_{10}$ onto *E. bicolor* metaphase chromosomes.

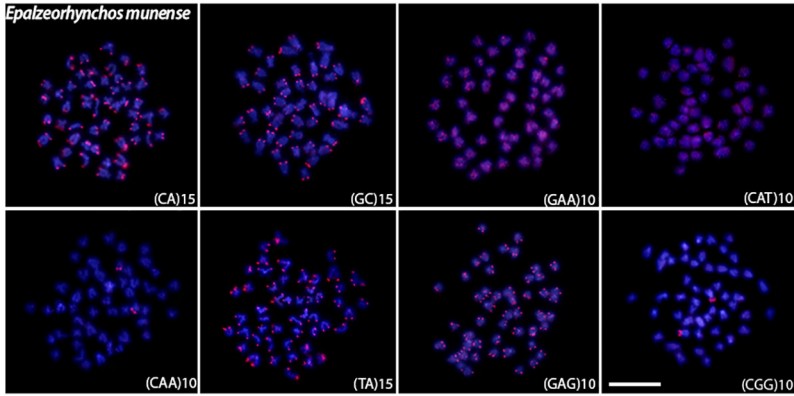

**Fig 7. Microsatellite sequences mapped onto *E. munense* chromosomes.** Distribution of microsatellite sequences $(CA)_{15}$, $(GC)_{15}$, $(TA)_{15}$, $(CAA)_{10}$ $(GAA)_{10}$ $(CAG)_{10}$ $(CAT)_{10}$ and $(CGG)_{10}$ onto *E. munense* metaphase chromosomes.

telomeric signals for the $(CAA)_{10}$ probe. The $(TA)_{15}$ was dispersed in *L. chrysophekadion* and *H. siamensis* chromosomes, but accumulated in telomeres of all other species. The $(CAG)_{10}$ was found dispersed in *L. chrysophekadion* and *T. thynnoides*, in the telomeres of all chromosomes of *E. munense*, restricted to the telomeric region of a single chromosome pair in *E. bicolor*, and bitelomeric in a single pair of *H. siamensis* in addition to a telomeric mark on short arm of a metacentric pair. Finally, in *H. siamensis* and *T. thynnoides*, the $(CGG)_{10}$ probe was found in the telomeres of all chromosomes, but in *E. munense*, it was only identified as a strong signal on one pair in the centromeric region, in contrast to a scattered pattern in *E. bicolor* and *T. thynnoides* chromosomes.

## Discussion

For the great majority of species reviewed in Table 1, including those in this study, Labeoninae has 2n = 50, a diploid number which is also hypothesized as a basal for cypriniform fishes [11, 18, 84]. Despite their conservative 2n, significant variations in karyotype structures were observed in both *Epalzeorhynchos* species as well as other ones investigated herein. The

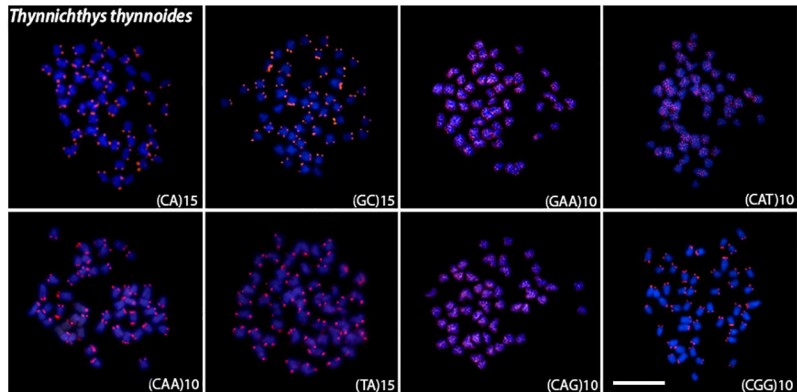

**Fig 8. Microsatellite sequences mapped onto *T. thynnoides* chromosomes.** Distribution of microsatellite sequences $(CA)_{15}$, $(GC)_{15}$, $(TA)_{15}$, $(CAA)_{10}$ $(GAA)_{10}$ $(CAG)_{10}$ $(CAT)_{10}$ and $(CGG)_{10}$ onto *T. thynnoides* metaphase chromosomes.

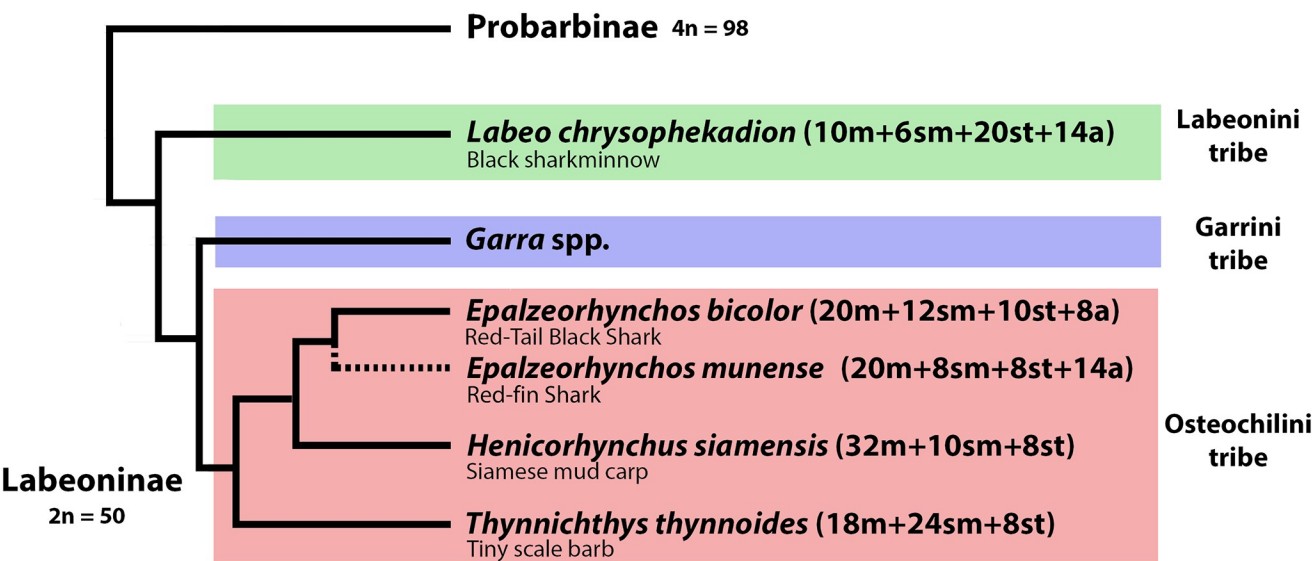

**Fig 9. Mapping of our cytogenetic results onto modified phylogenetic relationships of the Labeoninae based on nuclear and mitochondrial genes by Yang et al. [2] and classification of Tan and Armbruster [1].** The common names of the species are indicated under the scientific name. The position of *Epalzeorhynchos munense* was manually added to our figure, since it is not included in the [2] phylogeny, thus requiring confirmation.

different proportions of bi-armed chromosomes (metacentric and submetacentric) in the karyotypes imply that rearrangements like pericentric inversions may have acted during their karyotypic differentiation. It is also remarkable that a variety of karyotype re-organization events occured among the diversification of populations, as observed in the distribution of Ag-NOR/ 18S rDNA. Even though rDNAs are conservative elements of the eukaryotic genomes [85], the 18S rDNA typically occupies a terminal position in chromosomes, suggesting that these regions could represent recombination hotspots [86], feature also observed in Labeoninae karyotypes here investigated. In comparison, the 5S rDNA was found in both telomeric and centromeric regions, as also observed in the sister family ´Poropuntiinae´ (Cyprinidae). There, the 5S rDNA shows mostly a centromeric distribution, as observed in *Hypsibarbus* spp. and *Mystacoleucus* spp. [87]. Indeed, rDNA elements showed strikingly different patterns among cyprinid species, particularly in terms of variability in the number and position of 5S rDNA sites versus the more stable pattern of 18S rDNA sites. Such differences were especially notable in the sister clades *Epalzeorhynchos* and *Henichorhynchus*, where the former has a single chromosome pair carrying the 5S rDNA, while the last, four pairs. The mapping of repetitive sequences, especially ribosomal genes, has proven useful for estimating evolutionary karyotype changes in related species [86].

We demonstrated that the karyotype of *E. bicolor* is notably distinct from other populations described by Donsakul and Magtoon [44] (treated as *Labeo bicolor*). In their report, the karyotype contains a higher number of "a" chromosomes (12 pairs) than in our study (4 pairs). Such an enormous difference does not occur in the description of the karyotype of *E. munense*, where the acrocentrics vary from 6 to 8 pairs among populations. The extensive numerical variation of acrocentric chromosomes indicates that pericentric inversions might have occurred during the differentiation of their karyotypes. These rearrangements have been hypothesized to be crucial for local speciation and adaptation as well as for recombination suppression, especially in sex chromosomes [88–93]. This anticipated mechanism is widespread in cyprinoids chromosome evolution since most maintain the common 2n = 50, but are variable in the

karyotype composition. Both *Epalzeorhynchos* species also greatly differ in the distribution of microsatellite sequences, as for the $(GC)_{15}$, $(CAA)_{10}$, and $(CAG)_{10}$ microsatellites. Although mostly restricted to telomeric regions, where a significant fraction of repetitive DNA is localized [19], the number of chromosomes carrying these abovementioned motifs suggests that the dynamics of repetitive DNAs can be related to species divergences of this genus. In the genome evolution of various fish groups, the role of repetitive DNAs has been demonstrated [17–20, 94–98].

The proportion of biarmed chromosomes in Cyprinidae karyotypes is typically significantly greater than that of uniarmed ones [11]. The *Epalzeorhynchos* + *Henicorhynchus* + *Thynnichthys* clade (Osteochilinae) has a significantly higher number of (20) metacentric chromosomes than its sister clade Labeoninae, which possessed approximately 10 metacentric chromosomes making up their karyotypes as exemplified here in *L. chrysophekadion* (Fig 9). However, *Labeo bata* and *L. dero* have a more similar pattern to Osteochilini, with 18 and 26 metacentrics in karyotypes, respectively. The same higher proportion of metacentrics also occurs in Garrinae [4] and a high number of "m" chromosomes might be a plesiomorphic feature of the Osteochilinae and Garrinae, while the lower metacentric number in Labeoninae could represent a apomorphic feature of this tribe.

There are several fish species were 2n conservatism is found. Karyotype stasis/conservatism is primarily observed in marine fishes and is associated with a high degree of synteny between various DNA classes (e.g. Ellegren [99], Zhang et al. [100]). However, the conservation of 2n frequently follows the substantial changes in the karyotype structures and the distribution of repetitive DNAs in some families of Cyprinoidea. For example, the great 2n variation spanning from 42 in *Acheilognathus gracilis* (Acheilognathidae) [101] to 446 in *Ptychobarbus dipogon* (Cyprinidae) [102] has been recorded by families that experienced polyploidy throughout their chromosome evolution processes. In this context, the maintenance of 2n = 50 could represent the conservation of an ancestral trait in cyprinids [11, 84].

Our findings contributed to the knowledge of karyotypes and chromosome characteristics in the Labeoninae. Overall, these characteristics show that structural chromosome rearrangements, such as pericentric inversions, were mainly in charge of the wide numerical diversity of acrocentric chromosomes and had a major impact on the evolution of this cyprinid group. According to the detailed cytogenetic survey, trends of cytogenomic divergences can be identified in these Labeoninae species, which largely correspond to the inferred phylogenetic tree. Repetitive DNAs that demonstrated specificity in their distribution among species, such as ribosomal and microsatellite DNAs, were also effective indicators and drivers of particular genomic differentiation within Labeoninae.

## Author Contributions

**Conceptualization:** Sudarat Khensuwan, Francisco de Menezes Cavalcante Sassi, Thomas Liehr, Weerayuth Supiwong, Marcelo de Bello Cioffi.

**Data curation:** Francisco de Menezes Cavalcante Sassi, Renata Luiza Rosa de Moraes, Petr Rab, Nathpapat Tantisuwichwong, Marcelo de Bello Cioffi.

**Formal analysis:** Sudarat Khensuwan, Francisco de Menezes Cavalcante Sassi, Renata Luiza Rosa de Moraes.

**Funding acquisition:** Weerayuth Supiwong, Alongklod Tanomtong.

**Investigation:** Sudarat Khensuwan, Francisco de Menezes Cavalcante Sassi, Alongklod Tanomtong, Marcelo de Bello Cioffi.

**Methodology:** Sudarat Khensuwan, Francisco de Menezes Cavalcante Sassi, Renata Luiza Rosa de Moraes, Petr Rab, Thomas Liehr, Alongklod Tanomtong, Marcelo de Bello Cioffi.

**Project administration:** Weerayuth Supiwong, Kriengkrai Seetapan.

**Resources:** Weerayuth Supiwong.

**Supervision:** Thomas Liehr, Weerayuth Supiwong, Kriengkrai Seetapan, Marcelo de Bello Cioffi.

**Validation:** Petr Rab, Thomas Liehr, Weerayuth Supiwong, Kriengkrai Seetapan, Nathpapat Tantisuwichwong, Satit Arunsang, Phichaya Buasriyot, Sampun Tongnunui, Marcelo de Bello Cioffi.

**Visualization:** Sudarat Khensuwan, Petr Rab, Satit Arunsang, Phichaya Buasriyot, Sampun Tongnunui.

**Writing – original draft:** Sudarat Khensuwan, Francisco de Menezes Cavalcante Sassi, Renata Luiza Rosa de Moraes, Marcelo de Bello Cioffi.

**Writing – review & editing:** Sudarat Khensuwan, Francisco de Menezes Cavalcante Sassi, Renata Luiza Rosa de Moraes, Petr Rab, Thomas Liehr, Weerayuth Supiwong, Kriengkrai Seetapan, Alongklod Tanomtong, Nathpapat Tantisuwichwong, Satit Arunsang, Phichaya Buasriyot, Sampun Tongnunui, Marcelo de Bello Cioffi.

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
