## [Decision Letter · Decision Letter 0]

13 Jun 2023

PONE-D-23-14589Chromosomes of Asian cyprinid fishes: Novel insight into the chromosomal evolution of Labeoninae (Teleostei, Cyprinidae)PLOS ONE

Dear Dr. Supiwong,

Thank you for submitting your manuscript to PLOS ONE. After careful consideration, we feel that it has merit but does not fully meet PLOS ONE’s publication criteria as it currently stands. Therefore, we invite you to submit a revised version of the manuscript that addresses the points raised during the review process.

We look forward to receiving your revised manuscript.

Kind regards,

Ishtiyaq Ahmad, Ph.D

Academic Editor

PLOS ONE

Journal Requirements:

4. We noticed you have some minor occurrence of overlapping text with the following previous publication(s), which needs to be addressed:

https://www.scielo.br/j/gmb/a/FYWGsxSngHdjW9B7HWrBZfw/?lang=en

https://www.mdpi.com/2076-2615/13/8/1415

In your revision ensure you cite all your sources (including your own works), and quote or rephrase any duplicated text outside the methods section. Further consideration is dependent on these concerns being addressed.

5.In your Methods section, please provide additional information regarding the permits you obtained for the work. Please ensure you have included the full name of the authority that approved the field site access and, if no permits were required, a brief statement explaining why.

   "This research project is supported by the National Research Council of Thailand (NCRT):NRCT-RGJ63003-068. The Thailand science research and innovation fund and the University of Phayao (Grant No. FF66-UoE013)"

  "This research project is supported by the National Research Council of Thailand (NCRT): NRCT-RGJ63003-068. The Thailand science research and innovation fund and the University of Phayao (Grant No. FF66-UoE013)."

7. Thank you for stating the following financial disclosure: 

   "This research project is supported by the National Research Council of Thailand (NCRT): NRCT-RGJ63003-068. The Thailand science research and innovation fund and the University of Phayao (Grant No. FF66-UoE013)."

8. Your ethics statement should only appear in the Methods section of your manuscript. If your ethics statement is written in any section besides the Methods, please delete it from any other section. 

9. We note that Figure 1 in your submission contain [map/satellite] images which may be copyrighted. All PLOS content is published under the Creative Commons Attribution License (CC BY 4.0), which means that the manuscript, images, and Supporting Information files will be freely available online, and any third party is permitted to access, download, copy, distribute, and use these materials in any way, even commercially, with proper attribution. For these reasons, we cannot publish previously copyrighted maps or satellite images created using proprietary data, such as Google software (Google Maps, Street View, and Earth). For more information, see our copyright guidelines: http://journals.plos.org/plosone/s/licenses-and-copyright.

10. Please include a copy of Table 2 which you refer to in your text on page 4.

Reviewers' comments:

Reviewer's Responses to Questions

**Comments to the Author**

1. Is the manuscript technically sound, and do the data support the conclusions?

Reviewer #1: Partly

Reviewer #2: Yes

2. Has the statistical analysis been performed appropriately and rigorously? 

Reviewer #1: N/A

Reviewer #2: N/A

3. Have the authors made all data underlying the findings in their manuscript fully available?

Reviewer #1: Yes

Reviewer #2: Yes

4. Is the manuscript presented in an intelligible fashion and written in standard English?

Reviewer #1: No

Reviewer #2: No

5. Review Comments to the Author

Reviewer #1: The manuscript (PONE-D-23-14589) entitled ‘Chromosomes of Asian cyprinid fishes: Novel insight into the chromosomal evolution of Labeoninae (Teleostei, Cyprinidae)’ has been carefully reviewed as per the guidelines of the journal. In my point of view, the manuscript is judged to be of potential interest to the readers and the scientific community as well. While the overall concept of the paper is reasonably satisfactory, it falls within the scope of the journal. However, it has a number of flaws, which need to be addressed before it is formally accepted for publication. For instance, there are some errors in the manuscript's scientific, typographical, and grammatical content. Overall, this manuscript may offer important contributions to the literature, but needs some major corrections for possible publication in this journal.

See attachment

Reviewer #2: 1.There are typographical errors. Results should be in past tense and discussion should be in present tense.

2. More details should be added for molecular results in the discussion and should be compared with earlier work.

3. Explanation of figures should be added under the figures not in the text.

4. Results of molecular studies with different probes should be compared with respect to each species because all the results compiled in one figure for each species. So, it is difficult to compared the results.

5. Figure numbers for chromosome plates whether A/B/C should be added in the text.

6. There are 104 references but these are not properly discussed in the text.

7. The localization of 5S or 18S RNA should be marked on the plates with arrows to confirm their presence. same with the results of different probes.

8. The data is very less for phylogenetic analysis. more data should be added to draw evolutionary relationships based on phylogenetic tree to justify the title of the paper.

6. PLOS authors have the option to publish the peer review history of their article (what does this mean?). If published, this will include your full peer review and any attached files.

Reviewer #1: **Yes: **Azra Nabi Shah

Reviewer #2: No

---

## [Author Response · Author response to Decision Letter 0]

26 Jul 2023

June 22nd, 2023

Dear Editor of PLOS One,

We would like to submit the revised version of the manuscript entitled “Chromosomes of Asian cyprinid fishes: Novel insight into the chromosomal evolution of Labeoninae (Teleostei, Cyprinidae) to be considered for publication as an original article in PLOS One. 

An itemized list of all changes made, in response to comments from the Editor and reviewers, is given below and all the changes performed are highlighted as track changes in the revised manuscript (attached). We appreciate the contributions pointed out by reviewers and the editor. 

Editorial Comments

We modified the files according to the templates.

This information was added in the sampling subsection. 

We copyedit the entire manuscript and changes are indicated in the revised manuscript file.

4. We noticed you have some minor occurrences of overlapping text with the following previous publication(s), which needs to be addressed:

https://www.scielo.br/j/gmb/a/FYWGsxSngHdjW9B7HWrBZfw/?lang=en

https://www.mdpi.com/2076-2615/13/8/1415

In your revision, cite all your sources (including your own works), and quote or rephrase any duplicated text outside the methods section. A further consideration is dependent on these concerns being addressed.

We modified the manuscript to avoid overlapping with the abovementioned sections. 

5.In your Methods section, please provide additional information regarding the permits you obtained for the work. Please ensure you have included the full name of the authority that approved the field site access and, if no permits were required, a brief statement explaining why.

Such information is described in the sampling subsection. “All procedures followed the authorization of the Animal Ethics Committee of KhonKaen University based on the Ethics of Animal Experimentation of the National Research Council of Thailand (Record No. IACUC-KKU-105/63), the animal care and use license for scientific research was U1-07864-2561 (COA.No.MU-IACUC 2019/007) for the fishes of the Mae Klong Basin in Thailand approved from Mahidol University and fish collection authorized by Department of fisheries in Thailand, license number was 49/2564, respectively.”

 "This research project is supported by the National Research Council of Thailand (NCRT):NRCT-RGJ63003-068. The Thailand science research and innovation fund and the University of Phayao (Grant No. FF66-UoE013)"

 "This research project is supported by the National Research Council of Thailand (NCRT): NRCT-RGJ63003-068. The Thailand science research and innovation fund and the University of Phayao (Grant No. FF66-UoE013)."

We removed the Acknowledgements section in the revised manuscript. The correct funding information is the one indicated in the next question.

7. Thank you for stating the following financial disclosure: 

 "This research project is supported by the National Research Council of Thailand (NCRT): NRCT-RGJ63003-068. The Thailand science research and innovation fund and the University of Phayao (Grant No. FF66-UoE013)."

Role of Funder:

8. Your ethics statement should only appear in the Methods section of your manuscript. If your ethics statement is written in any section besides the Methods, please delete it from any other section. 

We removed this information and the whole Supporting Information section.

9. We note that Figure 1 in your submission contain [map/satellite] images which may be copyrighted. All PLOS content is published under the Creative Commons Attribution License (CC BY 4.0), which means that the manuscript, images, and Supporting Information files will be freely available online, and any third party is permitted to access, download, copy, distribute, and use these materials in any way, even commercially, with proper attribution. For these reasons, we cannot publish previously copyrighted maps or satellite images created using proprietary data, such as Google software (Google Maps, Street View, and Earth). For more information, see our copyright guidelines: http://journals.plos.org/plosone/s/licenses-and-copyright.

This map was created by our research group using only copyright-free data available on QGIS. 

10. Please include a copy of Table 2 which you refer to in your text on page 4.

Table included in the revised manuscript files.

Reviewer 1

The manuscript (PONE-D-23-14589) entitled ‘Chromosomes of Asian cyprinid fishes: Novel insight into the chromosomal evolution of Labeoninae (Teleostei, Cyprinidae)’ has been carefully reviewed as per the guidelines of the journal. In my point of view, the manuscript is judged to be of potential interest to the readers and the scientific community as well. While the overall concept of the paper is reasonably satisfactory, it falls within the scope of the journal. However, it has a number of flaws, which need to be addressed before it is formally accepted for publication. For instance, there are some errors in the manuscript's scientific, typographical, and grammatical content. Overall, this manuscript may offer important contributions to the literature, but needs some major corrections for possible publication in this journal.

Title

Title is long but informative.

Abstract

The abstract is concise, factual, and states briefly the purpose of the research. I would recommend suggesting some implications from the findings to represent future prospects of the subject matter. Besides, write full scientific name of E. munense as it is in first use.

We performed the modifications as suggested.

Introduction

The introduction part is poorly written. Rewrite the whole introduction part in order to meet its scientific clarity.

We have rewritten great part of the Introduction to make it more clear, following the reviewer request.

Materials and Methods

This section is also written very weak. I suggest consulting any English expert to improve its language quality.

We performed a grammar checking in the whole manuscript.

Results

All results are ok.

Discussion

Although discussion part is somehow linked to the results, but authors need to compare the present results with already published work. 

Our discussion section compared the herein obtained results with previous published works

Figures

Quality of figure 1, 2 and 3 is not acceptable as the quality of these figures is very poor. So, please revise these figures and change its dpi to 300. 

Modifications performed as suggested.

References

Crosscheck all references and arrange them in accordance with the guidelines of the journal. 

Moreover, the English language of the present version is not up-to-the-mark, and a lot of mistakes and syntaxes have been seen in the manuscript. It is firmly recommended to reconsider the English and get the manuscript to be checked by a professional editor

We performed a copyedit in the whole text to fit in the suggested criteria.

Reviewer 2

1.There are typographical errors. Results should be in past tense and discussion should be in present tense.

We checked the whole text in the revised version.

2. More details should be added for molecular results in the discussion and should be compared with earlier work.

We modified the discussion section to include such information.

3. Explanation of figures should be added under the figures not in the text.

We added those informations on the figure legends.

4. Results of molecular studies with different probes should be compared with respect to each species because all the results compiled in one figure for each species. So, it is difficult to compared the results.

We understand that a single figure comparing all species can be better to compare the results. However, we are afraid that chromosomes will be very small in the publication and no clear patterns will be observed. To solve this, we described better the general trends and arranged the figures to appear in sequence on the published version.

5. Figure numbers for chromosome plates whether A/B/C should be added in the text.

We included this information in the revised version.

6. There are 104 references but these are not properly discussed in the text.

We appreciate the comments but in fact we believe that all references are properly addressed in the main text, including the discussion section. Moreover, we re-analyzed this point in the revised version and also checked for possible mistakes in references.

7. The localization of 5S or 18S RNA should be marked on the plates with arrows to confirm their presence. same with the results of different probes.

We included arrows to clarify the FISH signals. On scattered patterns we choose to do not include because they can be difficult to properly address the position. 

8. The data is very less for phylogenetic analysis. more data should be added to draw evolutionary relationships based on phylogenetic tree to justify the title of the paper.

We understand that the title does not infer about phylogenetic tree. We discussed the chromosomal data in an evolutionary context, based on previous phylogenetic trees.

Thank you for your kind consideration of this paper.

Yours sincerely, 

Weerayuth Supiwong 

Corresponding author

---

## [Decision Letter · Decision Letter 1]

27 Sep 2023

Chromosomes of Asian cyprinid fishes: novel insight into the chromosomal evolution of Labeoninae (Teleostei, Cyprinidae)

PONE-D-23-14589R1

Dear Dr. Supiwong,

We’re pleased to inform you that your manuscript has been judged scientifically suitable for publication and will be formally accepted for publication once it meets all outstanding technical requirements.

Kind regards,

Ishtiyaq Ahmad, Ph.D

Academic Editor

PLOS ONE

Reviewers' comments:

Reviewer's Responses to Questions

**Comments to the Author**

1. If the authors have adequately addressed your comments raised in a previous round of review and you feel that this manuscript is now acceptable for publication, you may indicate that here to bypass the “Comments to the Author” section, enter your conflict of interest statement in the “Confidential to Editor” section, and submit your "Accept" recommendation.

Reviewer #2: (No Response)

2. Is the manuscript technically sound, and do the data support the conclusions?

Reviewer #2: Yes

3. Has the statistical analysis been performed appropriately and rigorously? 

Reviewer #2: N/A

4. Have the authors made all data underlying the findings in their manuscript fully available?

Reviewer #2: Yes

5. Is the manuscript presented in an intelligible fashion and written in standard English?

Reviewer #2: Yes

6. Review Comments to the Author

Reviewer #2: N/A

7. PLOS authors have the option to publish the peer review history of their article (what does this mean?). If published, this will include your full peer review and any attached files.

Reviewer #2: **Yes: **GURINDER KAUR WALIA

---

## [Editor Report · Acceptance letter]

31 Oct 2023

PONE-D-23-14589R1 

Chromosomes of Asian cyprinid fishes: novel insight into the chromosomal evolution of Labeoninae (Teleostei, Cyprinidae) 

Dear Dr. Supiwong:

I'm pleased to inform you that your manuscript has been deemed suitable for publication in PLOS ONE. Congratulations! Your manuscript is now with our production department. 

Kind regards, 

on behalf of

Dr. Ishtiyaq Ahmad 

Academic Editor

PLOS ONE